# What is Developmental Dyslexia?

**DOI:** 10.3390/brainsci8020026

**Published:** 2018-02-04

**Authors:** John Stein

**Affiliations:** Department Physiology, Anatomy & Genetics, University of Oxford, Oxford OX1 3PT, UK; john.stein@dpag.ox.ac.uk; Tel.: +44-8651-272552

**Keywords:** Dyslexia, reading, magnocellular neurons, vision, hearing, phonology, sequencing, timing, temporal processing, transient, coloured filters, rhythm, music, omega 3s

## Abstract

Until the 1950s, developmental dyslexia was defined as a hereditary visual disability, selectively affecting reading without compromising oral or non-verbal reasoning skills. This changed radically after the development of the phonological theory of dyslexia; this not only ruled out any role for visual processing in its aetiology, but it also cast doubt on the use of discrepancy between reading and reasoning skills as a criterion for diagnosing it. Here I argue that this theory is set at too high a cognitive level to be explanatory; we need to understand the pathophysiological visual and auditory mechanisms that cause children’s phonological problems. I discuss how the ‘magnocellular theory’ attempts to do this in terms of slowed and error prone temporal processing which leads to dyslexics’ defective visual and auditory sequencing when attempting to read. I attempt to deal with the criticisms of this theory and show how it leads to a number of successful ways of helping dyslexic children to overcome their reading difficulties.

## 1. Introduction

For most of the last century, the answer to the question ‘What is developmental Dyslexia?’ would have been simple. Developmental dyslexia was a hereditary deficit selectively affecting the visual processing of words, but leaving oral and non-verbal reasoning skills intact. The key features were a discrepancy between inability to acquire visual reading skills, yet normal oral and nonverbal abilities, and a genetic background.

In 1878, Adolf Kussmaul introduced the concept of ‘Word blindness’ to apply to stroke patients who had selectively lost their ability to read but had retained intact their oral and non-verbal reasoning skills; this was demonstrated by a great discrepancy between their inability to read and their preserved oral intelligence after the stroke. In 1884, Rudolph Berlin named this condition ‘dyslexia’ [1]. A few years later, Pringle Morgan described his famous case of ‘congenital word blindness’—Percy, a boy who had found it impossible to learn to read, despite having apparently very high oral and non-verbal intelligence [2]. We would now call this: developmental dyslexia. Morgan was convinced that Percy had a visual processing problem selectively targeting written words. He was also strongly of the opinion that it was ‘congenital’, i.e., due to a hereditary defect. These ideas were developed further in the first half of the 20th Century, in particular by Cyril Hinshelwood in the UK [3] and Samuel Orton in the USA [4]. Hence, until the mid-20th Century, most people thought that developmental dyslexia was a hereditary deficit selectively affecting the visual processing of words, leaving oral language and non-verbal reasoning skills relatively intact.

However, in the 1950s, Noam Chomsky introduced his revolutionary approach to linguistics by developing the principles of ‘generative phonology’ [5]. By the 1980s this had completely transformed views of developmental dyslexia as well. The hypothesis that it may have a visual basis was utterly rejected. From being regarded as an abnormality of the brain’s visual processing, developmental dyslexia was recast as a language disorder, a failure to acquire phonological skills [6]. This occurred despite the fact that many dyslexics appear to have no speech or language problems at all. Nevertheless, developmental dyslexia became the province of linguistic and educational psychologists, rather than neurologists.

## 2. The Phonological Theory

This phonological theory of developmental dyslexia postulates that dyslexic children fail to learn to read because they fail to acquire the skill of separating the sounds in a word to match with its visual letter counterparts. This is often known as ‘phonemic awareness’, which can conveniently be assessed by getting children to read pseudo-words (also known as non-words), e.g., ‘tegwop’, a word that can easily be read using letter/sound correspondence rules, but means nothing. Earlier in development, ‘phonological awareness’ begins to be acquired through practising rhyme, alliteration and other word games; these can also be used to measure it. The successful emergence of phonological awareness is a very strong predictor of later linguistic and reading competence [7]. Recent imaging studies have shown clearly that the phonological impairments in dyslexics are associated with significant abnormalities not only in cerebral connectivity, but also in cortical structure, particularly involving the left hemisphere language network [8,9]. These results show that the phonological theory is by no means completely erroneous, merely incomplete. 

## 3. Discrepancies

An unfortunate consequence of the dominance of the phonological theory is that it is now almost impossible to define dyslexia as a category of reading disability that can be distinguished from other possible reasons for reading failure, such as low general ability, poor teaching, family stress, lack of family support. This is because, whatever its cause, failure to learn to read always results from not being able to learn how to split words down into their separate sounds in order to match them with the letters that symbolize them—this is, after all, the very essence of reading. Hence, every child who fails to learn to read has phonological problems to some extent. In addition, so there are no differences between the phonological abilities of dyslexics compared with all the other children who fail to learn to read for other reasons [10]. In consequence, one cannot distinguish dyslexia from other causes of reading failure on phonological grounds alone. The same argument applies also to attempts to use the discrepancy criterion, described above, to identify dyslexia. Whether or not a child has a discrepancy between their oral abilities and their reading abilities, their phonological problems will be the same [10,11]. 

## 4. Response to Intervention

To attempt to overcome some of these problems with the phonological theory, another criterion is often added, namely response to intervention (RTI) [12]. The idea is that if one can identify a group of poor readers who respond well to well-founded phonological interventions, these are the children who might be called dyslexic [13]. However, this approach raises more questions than it answers, because there is no agreement about which interventions, nor which tests of success, to use, and expecting agreement between 2 correlated tests with even small amounts of unreliability is not realistic [14]. Furthermore, the argument is somewhat circular, because defining as dyslexic those who are poor on phonological tests and then improve with phonological training is starting with the assumption that you are trying to prove, namely, that poor phonological skills are the cause of dyslexia. This, of course, neglects any visual causes and also intractable auditory/phonological processing deficiencies that fail to respond to extensive phonological training. 

Those who actually assess and teach children with reading difficulties do routinely look for ‘spiky’ psychometric profiles, i.e., discrepancies between reading and oral or nonverbal skills, in order to identify dyslexia. Despite this, it is now seriously argued by some that it is completely pointless to try to diagnose dyslexia at all. Either everybody with reading problems is dyslexic, or nobody is [15]. This unsatisfactory state of affairs has resulted directly from the contemporary dominance of the phonological theory, because in effect the theory merely restates that the children cannot read. It is almost a tautology, because translating letters into the sounds they represent is the essence of what reading is. Therefore, the result of the current dominance of the phonological theory is that now we cannot state with any sort of precision what dyslexia actually is.

## 5. Pathophysiology

Despite its general acceptance, the phonological theory of dyslexia can hardly be called explanatory. What we need to find out is why children fail to gain phonological skills; what are the pathophysiological mechanisms which cause this failure. A much more useful theory would be one which properly explained why these problems arise, and hence enable us to define precisely what developmental dyslexia is. 

In this review, I propose that, for now, we return to the early 20th Century view that developmental dyslexia is the inability to learn to read despite normal oral and non-verbal abilities, with a strong hereditary (genetic) background. Hence, to diagnose dyslexia, we need to show a family history together with big differences between a child’s non-verbal and oral intelligence and his reading abilities—in short, a significant discrepancy. Fortunately, however, in the near future, recent advances in neuroscience should mean that much greater understanding of the causes of reading failure will emerge. So diagnosis should no longer depend on the demonstration of discrepancy between general oral and nonverbal reasoning abilities compared with very poor reading, which is at best only an indirect indicator of its underlying causes. Instead, we should be able to directly measure the deficient visual and auditory processing that causes the discrepancy in the first place, in order to make a diagnosis of dyslexia.

## 6. Sequencing

For it has become clear that the basic problem that dyslexics face is that they fail to learn to properly sequence the letters in a word and/or the sounds of which it is composed [16,17]. Prereading children destined to become dyslexic are slower and less accurate than their peers in recognizing individual letters; and poor letter recognition at the age of four or five is the strongest predictor of later problems with learning to read [18]. This slowness translates to slower visual recognition of the letters in whole words and, hence, problems with sequencing them. This poor sequencing turns out to be one of the greatest problems that these children face. However, slow sequencing is not just confined to the visual system—difficulty with seeing and remembering the order of the letters in a word—it also affects the auditory system. Many dyslexics have great difficulty hearing the sounds in a word in the correct order [19].

## 7. Temporal Processing

All these problems, slow recognition of letters and poor ability to sequence the order of letters and their sounds, derive from a basic impairment of sensory timing, of the visual inputs during reading and of the sounds in the word when hearing it; in other words, it is a basic problem with ‘temporal processing’ [20]. To visually sequence the letters in a word accurately we need first to focus attention on the first letter in order to identify it, then shift attention to the next letter while retaining in memory the position of the first, then shift to the next retaining the position of the first and second letters, and so on. Hence, accurate signalling of these shifts of attention by the visual system is essential [17]. In young children viewing large widely spaced letters, similarly accurate signalling of their eye movements is also essential. The crucial feature of these shift signals is their timing. If we tag the first letter we identify with the point in time at which we focused attention on it and the second letter with the time we focused on it and so on, we can easily recover the order in which we saw them all.

## 8. Magnocellular Neurons

This timing function depends particularly upon the properties of a specialised set of large neurones found throughout the brain; they are often called magnocellular neurones (L. magnus = large) because of their size. The larger a neurone is, the faster are all the processes involved in its transmission of signals. In the case of the visual system, the visual magnocellular neurones provide this rapid processing. These neurones are specialised not for defining the fine detail or colour of letters, but for timing when the visual system first senses one, which can direct visual attention to focus on it, and then direct an eye movement to position the letter on the fovea, the part of the retina with the highest acuity, in order to identify it.

10 percent of the retinal ganglion cells are these large magnocells. They collect inputs from up to 50 times the area covered by the smaller parvo cells [21]. Their large size means that they cannot resolve fine detail, e.g., the difference between an ‘a’ and an ‘o’; but they can respond much faster. They receive mainly from the long wavelength (red and green) cones and also from the rods, the receptors that respond at very low light levels. Thus, they are most sensitive to yellow light at low intensities. They do not contribute to conscious colour vision, but only to the timing of light inputs. They are therefore particularly sensitive to movement. They signal when something happens, but only indicate roughly where it is located spatially, so they are often called ‘blob’ detectors. 

The axons of the magnocells project to the magnocellular layers of the lateral geniculate nucleus (LGN), which is a relay in the thalamus that projects onwards to Layer 4C alpha in the primary visual cortex (V1 or striate cortex), which is situated at the back of the occipital lobe. The other 90 percent of retinal ganglion cells (mainly parvocells and koniocells) are responsible for detecting the fine detail and colour of an object in order to identify it. These project to the parvocellular layers of the LGN, thence to layer 4C beta in the striate cortex.

## 9. Dorsal and Ventral Visual Cortical Networks

After this, the magnocellular and parvocellular layers interact strongly. However, two main processing streams pass forwards from the primary visual cortex in each hemisphere, the so-called dorsal and ventral striatofugal pathways [22]. The dorsal of these is dominated by its magnocellular input. 90 percent is magnocellular; only 10 percent parvocellular. It projects via the middle temporal or ‘motion’ area to the posterior parietal cortex, and thence to the prefrontal cortex. Thus, it mediates the visual direction of attention and the visual guidance of eye and other body movements. It is, therefore, often called the ‘Where’ stream. 

The ventral pathway passes to the inferotemporal cortex. Often termed the ‘What’ stream, it is responsible for the identification of objects. Hence, the ‘visual word form area’ (VWFA), which is central to reading [23], is found on this pathway in the left inferotemporal (fusiform) cortex. However, it also receives a large input from the dorsal pathway, which is believed to direct visual attention to whatever letter currently needs to be identified [24].

## 10. Impaired Magnocellular Development

In the context of dyslexia, the reason for discussing these magnocellular systems at all is that there is now overwhelming evidence that in dyslexia their development is impaired. Only in the retina and the lateral geniculate nucleus are the magno and parvo systems anatomically completely separate, so strictly speaking, we can only identify the magnocellular system in those regions [25]. Many studies have now confirmed that in dyslexics, the magnocellular responses are reduced in both the retina [26] and the LGN. Indeed, it was the histological study of the LGN in brains donated to the Orton Society by some of Samuel Orton’s dyslexic patients, that first showed that their magnocellular layers were abnormal [27]. Modern neuroimaging techniques are now so sensitive that these LGN abnormalities have now been confirmed in life [28,29].

## 11. Impaired Dorsal Stream Function?

Although the dorsal stream is dominated by its magnocellular input we cannot equate the dorsal stream with the visual magnocellular system because it also receives a small parvocellular input [30]. However, by exploiting the high contrast sensitivity of magnocells at low luminances and high temporal frequencies, to which parvo cells do not respond at all, we can selectively stimulate the subcortical magnocellular system in order to decide whether its impairment impacts on dorsal stream function as a whole. This strategy has been successfully employed in a wide variety of psychophysical, electrophysiological, functional imaging and visuomotor experiments [31,32,33]. Most have concluded that most people with dyslexia show clear evidence of impaired function of the dorsal stream, and hence of abnormal visual magnocellular function.

## 12. Visual Attention

Since the dorsal stream plays such a crucial role in the control of the focus of visual attention and of eye movements for reading, we will consider some of these in more detail. In so-called ‘priming’ experiments, an arrow pointing to where a target is going to appear, is flashed briefly and this shortens the reaction time to the target when it does appear. This priming effect is much reduced in dyslexics because their sluggish M-responses slow their ability to shift their attention to the primed location [34]. 

Another way of testing visual attention is ‘visual search’. If one looks at an array of similar objects, one with a single difference from the others ‘pops out’ immediately (parallel search), but if two conjunctive features are required to distinguish it, it does not stand out, and can only be identified by searching through all of them systematically (serial search). Serial search requires sequential allocation of attention mediated by the dorsal stream, hence dyslexics have repeatedly been shown to be worse at it [17].

## 13. Eye Movement Control

Visual magnocells mediate the control of eye movements in several ways. For smooth pursuit they signal the speed of the movement of a target and thus enable the eyes to follow it. Slow motion processing in dyslexics therefore causes the eyes to fail to keep up with it, so their pursuit eye movements fall further and further behind, necessitating frequent catch up saccades, often called ‘saccadic intrusions’, which keep the eyes up with the target [35]. For fixation, they signal any ‘slip’ of the image being fixated; this is fed back to the eye muscle control centres to bring the eyes back on target. Slow feedback of this retinal slip when the eyes inadvertently lose fixation, mean that dyslexics’ fixation on letters when reading is very unstable [36,37]. For saccadic control magnocells detect the appearance of a new target and its location, which is used to program the saccade to fixate it. Additionally, during the saccade, the M-system signals the motion of images across the retina; this is then integrated to indicate the amplitude of the actually executed saccade. Thus, the impaired development of M cells in dyslexics causes impaired fixation stability, impaired smooth pursuit and inaccurate saccades. These problems are particularly compounded when vergence adjustments are needed, causing saccades from word to word in dyslexics to be characterised by large vergence errors [38]. 

Hence dyslexics with impaired M-function show longer fixations with a tendency for the letters to appear to move around, inaccurate and divergent saccades, and also eye strain with excessive blinking and headaches. However, many people attribute these symptoms to the fact that the children were having problems with their reading, rather than being the cause of them. Even though there have been many studies showing that dyslexics’ eye movements are abnormal even when tested with non-verbal stimuli, the consensus remains that any eye movement problems seen in dyslexics are the result, not the cause of their reading problems [39]. This needs to change.

## 14. Visual Treatments

Perhaps the most powerful way to show that the development of the visual magnocellular system is impaired would be to be able to demonstrate that improving its performance in dyslexics improved their reading. Lawton trained dyslexics’ magnocellular function by asking them to decide the direction of motion of progressively dimmer moving gratings on a background of similar high contrast gratings. As their contrast sensitivity improved, she reduced the contrast of the gratings, so that magnocellular sensitivity progressively increased, and with it the children’s reading [40]. Similarly, Chouake et al. [41] showed that training the magnocellular pathway to detect progressively faster movements was followed by improved lexical decision and reading accuracy. In another study, visuomotor training was carried out in Chinese children with dyslexia. The interventions consisted of progressively more difficult coherent motion detection, visual search, visual tracking and juggling. These exercises improved magnocellular pathway function, and this was associated with significant increases in phonological awareness [42]. Likewise, Gori et al. [43] showed that getting dyslexic children to play action video games with no reading or phonological content helped them to greatly improve both their visual magnocellular function and their reading. 

In another kind of study, participants’ eye movements were trained. Leong et al. gave elementary students saccadic training, and this improved their reading fluency significantly [44]. This has also been shown for a quite different language system, namely Persian. Eye movement training to improve vergence and accommodation led both to increments in reading comprehension and decrements in errors. Thus, it is clear that improving magnocellular function can improve reading. This supports the conclusions that sensitive visual magnocellular function is essential for learning to read and that these causal effects can be seen in all scripts, whether alphabetic or logographic, Roman, Farsi, Arabic, Kanji, Kana or Chinese.

## 15. Yellow Filters 

The possible benefits of viewing text through coloured filters is an extremely controversial subject [45,46,47,48,49,50,51]. Most of the studies have been small and badly designed without proper controls. Furthermore, they have lacked a plausible rationale, so that they have failed to attract much support. Nevertheless, most of those who actually use them to help children with visual reading problems, are certain that they work well in some children. Even so, the majority of researchers remain unconvinced [52,53,54]. 

However, the magnocellular theory does provide a potential rationale for using certain coloured filters. Although magnocells do not contribute to colour vision, they receive mainly from the long wave length cones, red and green. This means that they are best stimulated by yellow light, which consists of a mixture of red and green light. Skiers and sharpshooters know that wearing yellow goggles can improve their contrast sensitivity in ‘white out’ conditions [55]. This is because viewing through yellow filters causes the pupils to dilate. Hence the amount of yellow light entering the eyes increases at the expense of blue, thus facilitating the dorsal/magnocellular stream in the cerebral cortex. 

We have shown that having some children view text through such yellow filters can help many to improve their reading [56]. For a randomised control trial, we selected children who had shown improvements when looking at text through deep yellow overlays. Half, selected at random, were given, for three months, yellow filter spectacles to wear for all reading and close work. These improved their motion sensitivity significantly, and this was accompanied by marked improvements in their reading. The control children were given a card with a slit cut into it, in order to read words one at a time, a technique that had been shown to help some children [57], but this hardly improved our control children’s reading at all over the three months. In a much larger audit, we showed that such children could improve their reading by over six months in three months, and that this improvement persisted six months later [58]. Thus, it seems that viewing text through yellow filters can help some children to improve their magnocellular function, and hence their reading.

## 16. Blue Filters

However, we and others have found that another set of children with visual reading problems seem to do much better with dark blue filters, almost the opposite of the yellow ones [59]. These are children who complain that letters and words seem to move around and who also often complain of eye strain and headaches when they were trying to read. The blue filters probably help these children in a totally different way. Recently, a new kind of retinal ganglion cell has been discovered (the melanopsin containing ganglion cell) [60]. It does not contribute much to conscious vision; instead, it projects to the suprachiasmatic nucleus (SCN) in the hypothalamus. This nucleus is the brain’s central clock, which times out a 24-h period for regulating our many diurnal rhythms—sleep, blood pressure, body temperature, etc. However, day length varies between summer and winter, and therefore the SCN clock needs to be synchronised with prevailing day length [61]. This is the function of the melanopsin-containing ganglion cells. They are sensitive to blue light because blue light is the first to appear at daybreak; hence, this system arouses us at sunrise. 

Thus, when we give children blue glasses, we appear to be stimulating their arousal system by activating these melanopsin cells. In the audit referred to earlier [58], we found that giving appropriate children deep blue filters could improve their attention and improve their reading by more than nine months in three months. In addition, they improved the children’s sleeping patterns, because they helped synchronise this hypothalamic clock. Also, they seemed to improve the children’s headaches, probably by the same mechanism. Thus, blue filters can often help appropriate children by activating the melanopsin system to improve attention and arousal.

## 17. Criticisms of the Visual Magnocellular Hypothesis

Given the amount of evidence that has accumulated in favour of a visual magnocellular defect in dyslexics, it may seem surprising that it is not more generally accepted. However, wrongly, many people seem to presume that such an abnormality would be incompatible with the more popular phonological theory, even though the visual magnocellular theory actually offers a partial explanation for children’s phonological problems. For children only learn that words can be split down to their constituent phonemes after they have learnt that the word can be visually represented by the separate letters that stand for them [62].

These criticisms of the visual magnocellular theory take three main forms. First, it is pointed out that many dyslexics do not seem to have visual problems, and conversely also, that many children seem to be able to learn to read, despite having poor magnocellular function [63]. Thus, deficient magnocellular function cannot be considered either a necessary or sufficient condition for dyslexia. However, individuals’ visual motion sensitivity, which is taken as an index of magnocellular function, does predict their eventual reading ability quite powerfully [64]. In other words, visual magnocellular function is clearly a quantitative determinant of reading ability, but only one among many. As we shall see, auditory processing deficiencies are common as well.

Another frequent criticism is that the dorsal attentional and visuomotor pathway is not exclusively magnocellular [65]. This is true; 10 percent of its visual input is provided by the parvocellular system. However, many of the neurons of which the dorsal route is composed express the same surface ‘signature’ antigen as the magnocellular ganglion cells in the retina, indicating that they are of the same lineage, i.e., they form part of the same system. This signature antigen can be demonstrated in the dorsal stream using specific ‘magnocellular’ antibodies, such as CAT 301 [66]. Additionally, as we have seen, any contribution to the dorsal stream from the parvo system can be minimised by using stimuli that are of low contrast and high temporal frequency to which the parvo cells are blind. In many of the studies discussed above, such stimuli were used and the magnocellular deficit confirmed. 

A third kind of criticism is the argument that the magnocellular deficit may be a consequence, rather than a cause of reading impairment. Most of the studies discussed so far are correlational, which means that cause and effect cannot be established with certainty. However, in some studies, a ‘reading age match’ has been employed. In this design the dyslexic children are matched with a set of younger children who have the same reading age as the dyslexics. This means that their exposure to print will have been the same as that of the older dyslexics. Hence, if they do not display a magnocellular deficit, yet the dyslexics do, this is strong evidence that in the dyslexics, it is the magnocellular abnormality that has caused their reading problems rather than vice versa; their lack of reading experience could not have caused their magnocellular deficit [43]. 

However, a much-quoted fMRI study came to the opposite conclusion. Olulade et al. compared only 12 older dyslexic readers with 12 younger good readers, matched for reading age with the dyslexics [67]. They found that both groups had equivalent fMRI activity in V5/MT in response to a visual motion stimulus. However, many others using far larger numbers have shown that younger reading-age-matched children usually have better motion sensitivity than older dyslexics [68]. However, even if the younger good readers’ motion sensitivity was similar to that of the dyslexics, this shows no more than that even if you are 10 but you only have the motion sensitivity of a 7.5 year old, then you will not be able to read better than a 7.5 year old, i.e., it does not speak to what is cause and which is effect. They claimed that a ‘phonics’ intervention did in fact improve the dyslexic children’s fMRI visual motion responses somewhat, but actually the intervention had a strong visual component. Anyway, little can really be concluded from a study with such small numbers.

Another way of establishing cause and effect is by using a ‘cohort study’ in which a group of children is followed from prereading through school to see how their reading develops. Several such studies have demonstrated that the children with impaired magnocellular function in kindergarten are the ones who later develop reading problems [18]. However, as we have seen, the most convincing way of showing cause and effect is to demonstrate that improving magnocellular function can improve children’s reading progress, and this has been demonstrated in the numerous studies discussed earlier.

## 18. Auditory Temporal Processing

Although there is now overwhelming evidence that children with visual symptoms that impair their reading have a visual magnocellular deficit, some children with dyslexia do not appear to experience any visual problems at all; their difficulties appear to be mainly auditory. Not only does reading require visual sequencing of the letters in the written word, but also auditory sequencing of the phonemes within the spoken word. We distinguish these phonemes from each other by sensing the changes in sound frequency and amplitude that characterize them—their so-called frequency and amplitude modulations (FM & AM). These are detected by a specialised system of auditory transient neurons, which are large cells that stain for CAT 301 [69]. So they are in many ways analogous to the visual magnocellular cells, and some people refer to them as ‘auditory magnocellular’ neurones. 

We and others have shown that people with dyslexia are less sensitive to AM/FM modulations [70], and histological examination of large cells in the left auditory thalamic nucleus, the medial geniculate nucleus (MGN), which is equivalent to the visual LGN, have likewise been shown to be smaller in dyslexics [71]. Moreover, each individual’s sensitivity to frequency and amplitude modulations predicts his/her ability to do phonological tasks [72]. Hence, the sensitivity of many people with dyslexia to frequency and amplitude modulations is significantly reduced. Several groups have also shown that musical or rhythm training that improves dyslexics’ FM & AM sensitivity can help them to improve their phonological abilities [73,74]. 

This all goes to show that, like visual magnocellular impairment, auditory magnocellular impairment is an important cause of phonological failure in dyslexic children. In both children and adults, sensitivity to auditory modulations correlates quite strongly with their visual magnocellular sensitivity, suggesting that both are under similar genetic and environmental controls [64]. Since magnocellular dysfunctions can potentially be diagnosed much earlier than reading failure itself and any deficits will predict future reading problems, we should try to begin testing for visual and auditory temporal processing problems in children, long before they begin to fail to read, so that we can begin treatment and thus avoid the often tragic emotional and psychological consequences of that failure.

## 19. Magnocellular Neurones

In summary, temporal processing throughout the brain seems to be mediated by networks of magnocellular neurons, specialised for timing both internal and external events. They track changes (‘transients’) in light, sound, position, etc., for the direction of attention and the control of movements, hence for visual and auditory sequencing. They are found throughout the whole brain; in the visual system, the auditory system, the touch and proprioceptive systems, in the cerebral cortex, hippocampus, cerebellum and brainstem. Their size enables their rapid processing and transmission of signals about timing. 

All magnocells seem to derive from the same lineage, as they all express the same surface antigens, such as CAT 301 [75]. They are also very vulnerable. Impaired magnocellular development has been found in premature babies, in foetal alcohol syndrome, developmental dyslexia, dyspraxia, dysphasia, ADHD (), ASD, Williams syndrome, and even in schizophrenia and bipolar depression [76].

## 20. Omega 3 Fatty Acids

The high dynamic sensitivity of magnocellular neurons necessitates high membrane flexibility so that their ionic channels can open and close very fast. This flexibility is provided by their local lipid environment, particularly by the incorporation into the membrane of one very important omega 3 long chain fatty acid, Docosahexaenoic acid (DHA) [77,78]. This provides exactly the right physical and electrostatic properties for their membranes to allow optimum dynamic performance of the ionic channels passing through them [79]. 

DHA is normally provided in our diet by consuming oily fish. Green vegetables, flax or rape seed and seaweed all contain the shorter chain omega 3, alpha linolenic acid, but humans do not convert this into DHA very efficiently. As a result of their size and rapid temporal processing functions, magnocells seem to be particularly vulnerable to lack of DHA. This has become quite common in developed countries since oily fish, the main source of DHA for humans, ceased to be a significant component of our staple diets. Dyslexic children tend to have very low levels of DHA in their blood, and the degree of their blood DHA deficit predicts the magnitude of their reading deficit. Hence, we have found that supplementing the diet of dyslexic children with DHA can often help them to improve their reading, sometimes dramatically [80,81,82]. These observations all reaffirm the importance of magnocellular function for reading.

## 21. Conclusions

Thus, the answer to my original question ‘What is Developmental Dyslexia?’ is that it is a hereditary temporal processing defect, associated with impaired magnocellular neuronal development, that impacts selectively on the ability to learn to read, leaving oral and non-verbal reasoning powers intact. Armed with this definition, we should soon be able to test children specifically for these visual and auditory temporal processing deficits. This will not only enable us to diagnose dyslexia earlier, but also to set in motion remediation programmes tailored to each child’s particular, individual, pattern of needs.

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
