# Peer review of "What is Developmental Dyslexia?"

_brainsci, 2018, doi:10.3390/brainsci8020026_

Round 1

Reviewer 1 Report

The current study is a critical review and position paper advocating for the inadequacy of the Phonological Deficit Hypothesis (PDH) and in favor of the Magnocellular Deficit Hypothesis (MDH) of Dyslexia. The article is well-written, with a deliberate attempt to make difficult terms and concepts accessible for a generalist audience.

The introduction makes the point that the PDH is limited in that it makes Dyslexia difficult to differentially diagnose from other reading problems. I think that this is true, but that this point needs clarification and elaboration. The PDH is currently partnered with the Response to Intervention paradigm, as championed by Fletcher (see, for instance,Fletcher, J. M., Stuebing, K. K., Barth, A. E., Miciak, J., Francis, D. J., & Denton, C. A. (2014). AGREEMENT AND COVERAGE OF INDICATORS OF RESPONSE TO INTERVENTION: A MULTI-METHOD COMPARISON AND SIMULATION. Topics in Language Disorders, 34(1), 74–89. https://doi.org/10.1097/TLD.0000000000000004)

RTI is fundamentally a de-emphasis of etiology, and a de-medicalizing of school-related problems, and as such I see it more as a political position than a scientific one. What I would like to see a little more of is a discussion of the fact that the PDH has a basis in the behavioral and neuroimaging literature, and as such is just an incomplete theory, not an inherently problematic one. 

The neurophysiology and anatomy section is accurate and clear.

In establishing a link between clinical symptoms of RD/Dyslexia and MD', please see 

Desmond, A. M., Shelley-Tremblay, J., Tannen, B. M., Ciuffreda, K. J., & Larson, S. (2015). Correlation of Magnocellular Function with Measurements of Reading in Children. Vision Development and Rehabilitation1(2), 109–119.

Line 230, I would prefer not have have personal communications in peer-reviewed articles, perhaps there is a similar reference that has been published?

There is a newer Lawton reference that I can recommend, which includes a normal control 

Lawton, T., & Shelley-Tremblay, J. (2017). Training on Movement Figure-Ground Discrimination Remediates Low-Level Visual Timing Deficits in the Dorsal Stream, Improving High-Level Cognitive Functioning, Including Attention, Reading Fluency, and Working Memory. Frontiers in Human Neuroscience, 11. https://doi.org/10.3389/fnhum.2017.00236.

Section 12 on Visual Treatments could be strengthened by the inclusion of several studies from the labs at SUNY Optometry, including any of the following training studies:

Solan, H. A., Shelley-Tremblay, J. F., Hansen, P. C., & Larson, S. (2007). Is there a common linkage among reading comprehension, visual attention, and magnocellular processing? Journal of Learning Disabilities, 40(3), 270–278.

Solan, H. A., Shelley-Tremblay, J., Ficarra, A., Silverman, M., & Larson, S. (2003). Effect of Attention Therapy on Reading Comprehension. Journal of Learning Disabilities, 36(6), 556–563. https://doi.org/10.1177/00222194030360060601

Solan, H. A., Shelley-Tremblay, J., Hansen, P. C., Silverman, M. E., Larsone, S., & Ficarra, A. (2004). M-cell deficit and reading disability: a preliminary study of the effects of temporal vision-processing therapy. Optometry-Journal of the American Optometric Association, 75(10), 640–650.

Solan, H. A., Shelley-Tremblay, J., Larson, S., & Mounts, J. (2006). Silent Word Reading Fluency and Temporal Vision Processing Differences Between Good and Poor Readers. Journal of Behavioral Optometry, 17(6), 1–9.

Under the sections on Yellow and Blue filters, the author refers to individual studies, performed by himself. I would suggest that it would strengthen these sections to discuss meta-analyses or other reviews of the evidence, as mentioning 1 or 2 studies in isolation does not convince the reader. Indeed the cited studies are current and relevant, but more should be added.

In the section on Criticisms, the author reviews "Gori, S.; Seitz, A. R.; Ronconi, L.; Franceschini, S.; Facoetti, A." as evidence in favor of the causal role of MD's in dyslexia. The evidence from this study is clear, but see alternately: 

Olulade, O. A., Napoliello, E. M., & Eden, G. F. (2013). Abnormal Visual Motion Processing Is Not a Cause of Dyslexia. Neuron, 79(1), 180–190. https://doi.org/10.1016/j.neuron.2013.05.002

The findings of this study must be compared to those of the Gori study. The later study is an fMRI study, with its own limitations, but indeed it also cites several studies that find no behavioral evidence for m-cell deficit symptoms when controlling using age matching.

Overall, the need for such a position paper in the literature is high, with alternative explanations that add breadth to the understanding of reading problems. I recommend that whenever possible, the author provide additional references in favor his points. The article is "preaching to the choir" in terms of devotees to the MDT, but indeed is not likely to sway the educators and pediatricians who are taken with the PDT already without some additional evidence.

Author Response

Thank you for your very useful suggestions which I've endeavoured to incorporate.

'The PDH is currently partnered with the Response to Intervention paradigm' -

I've attempted to cover RTI, Fletcher et al., the imaging support for the PDH and PDH as not incorrect but incomplete in rewritten Sections 2-5.

I've added most of the refs. suggested in order to attempt not to overemphasise my own work!

I've discussed the controversies about use of coloured filters in more detail in sections 15 -16.

l. 249 reference to personal communication removed.

l. 334 Olulade et al. paper criticised in more detail.

I hope you will be satisfied with these changes and that they may convert some readers to join my choir!

Reviewer 2 Report

In this article the author proposes an interesting review, regarding a different point of view on developmental dyslexia. The magnocellular theory is one of the most controversial theories concerning developmental dyslexia, often criticized but among the few with specific neurophysiological basis. 

To increase the interest and comprehensibility, I think some small changes/comments could be added:

Chapter 3: Discrepancies (line 58): I think that this is the core of the article, it is essential to understand the underlying logic of the entire article (summarized in the phrase "the theory is set at too high a cognitive level". I think the addition of a small part about the limits of the phonological theory ("can hardly be called explenatory") should increase the interes for an alternative explanation.

Chapter 12 Visual treatments (line 213): Also in Gori et al. (2016; already cited [47], it is reported a magnocellular training similar to the ones of Lavidor or Chouake, that should be reported in the initial sequence along with other similar training.

Colored filters: in a recent meta-analysis, Galuschka, Krick & Schulte-Korne (2014) indicate that colored filters application did not result in an efficient training / instrument for children with dylslexia. a comment from the author on the controversial rendults could be interesting.

Author Response

Thank you for your very useful suggestions

Sections 2-5. Discrepancies and the limits of the phonological theory. I agree that this is the core of the article. I hope you will find the rewrite meets both reviewers' suggestions.

line 242. Gori et al. referenced.

section 15. More discussion of the controversies surrounding the use of coloured filters.